# Evolutionary Invariant of the Structure of DNA Double Helix in RNAP II Core Promoters

**DOI:** 10.3390/ijms231810873

**Published:** 2022-09-17

**Authors:** Anastasia V. Melikhova, Anastasia A. Anashkina, Irina A. Il’icheva

**Affiliations:** V.A. Engelhardt Institute of Molecular Biology, Russian Academy of Sciences, 119991 Moscow, Russia

**Keywords:** eukaryotes, transcription apparatus, core promoter DNA local structure, ultrasonic cleavage, DNase I cleavage, evolutionary invariant local structure of RNAP II core promoter

## Abstract

Eukaryotic and archaeal RNA polymerase II (POL II) machinery is highly conserved, regardless of the extreme changes in promoter sequences in different organisms. The goal of our work is to find the cause of this conservatism. The representative sets of aligned promoter sequences of fifteen organisms belonging to different evolutional stages were studied. Their textual profiles, as well as profiles of the indexes that characterize the secondary structure and the mechanical and physicochemical properties, were analyzed. The evolutionarily stable, extremely heterogeneous special secondary structure of POL II core promoters was revealed, which includes two singular regions—hexanucleotide “INR” around TSS and octanucleotide “TATA element” of about −28 bp upstream. Such structures may have developed at some stage of evolution. It turned out to be so well matched for the pre-initiation complex formation and the subsequent initiation of transcription for POL II machinery that in the course of evolution there were selected only those nucleotide sequences that were able to reproduce these structural properties. The individual features of specific sequences representing the singular region of the promoter of each gene can affect the kinetics of DNA-protein complex formation and facilitate strand separation in double-stranded DNA at the TSS position.

## 1. Introduction

The heterogeneity of the three-dimensional structure of the double-stranded DNA plays an important role in the regulation of genetic processes. This heterogeneity is modulated by the nucleotide sequence. Proteins may recognize the shape of DNA (“indirect readout”) or the unique chemical signatures of the DNA bases (“direct readout”) [1]. As a rule, DNA-binding proteins combine both readout mechanisms to achieve DNA-binding specificity [2].

RNA polymerase II (Pol II) in eukaryotes is responsible for the transcription of messenger RNA and some non-protein-coding small nuclear RNAs. Pol II core promoters are fragments of genomic DNA, about 100 bp long, surrounding the transcription start site (TSS). Transcription initiation occurs when TATA-binding protein (TBP) binds to the eight base-pair TATA elements of Pol II core promoter, coordinating accretion of class II initiation factors and Pol II into a functional preinitiation complex (PIC). This process is a slow stage of transcription; it leads to the formation of a long-lived protein-DNA complex [3].

The mechanical, thermodynamic, and structural properties of Pol II promoter regions have long attracted the attention of researchers [4,5,6,7,8,9]. Regardless of the length of the analyzed promoter fragment and analysis methods, all studies come to the same conclusion. In the vicinity of the TSS, all structural properties of DNA noticeably deviate from the average level, and core promoter regions are exceptionally heterogeneous.

The nucleotide sequences of the core promoters are usually represented by the DNA coding strand (namely, the strand with the 5′→3′ vector directed to the TSS from the upstream region; hereinafter, we will call it the upper strand). The TSS position is taken as coordinates −1, +1 (there is no nucleotide with zero coordinates). In all organisms, positions −2, +4 are occupied by the initiator element (INR). At this region, the complementary strands of the double helix diverge, and Pol II recognizes the template strand. TATA element in the promoters of most organisms is located at a distance of about −28 bp from the TSS.

The common regularities of the core promoter architecture in each species may be revealed after the superposition of signals from a huge amount of species’ promoter sequences properly aligned at the TSS. The well-annotated database of promoter sequences is an essential basis for identifying general patterns in the promoter structure. To analyze structural features of DNA that determine RNA polymerase II core promoter [10], we previously used the EPD New database [11]. The profiles of the averaged textual, structural, mechanical, and physicochemical characteristics in each position of the sets of 60 bp core promoter sequences (positions from −50 to +10) in the eight organisms available at that time from the EPD New database [11] (*H. sapiens, M. musculus, D. melanogaster, D. rerio, C. elegans, A. thaliana*, *S. cerevisiae*, *S. pombe*), were constructed. The analysis of these profiles allowed us to reveal the common scheme of the animal and plant core promoter architecture. The promoters of the unicellular fungus *S. pombe* were found to correspond to the same structural scheme, but the structure of the core promoter of another unicellular fungus, *S. cerevisiae*, turned out to be different [10].

To date, the number of organisms available for analysis in the EPD New database [12] has increased markedly. In addition to representatives of the Metazoa (vertebrates and invertebrates), plants, and unicellular fungi (*S. cerevisiae, S. pombe*), a representative of the Protozoa appeared, namely the parasite *P. falciparum*, whose genome is 80% AT-pairs. Moreover, the total number of promoters in the samples of those organisms that were previously represented in this database also increased noticeably. Therefore, it became possible to check the generality of the conclusions obtained by us earlier and to analyze the degree of influence of the percentage of AT pairs in the genomes of different organisms on the structural features of their promoters.

## 2. Results

The sets of promoters of fifteen evolutionarily different organisms were retrieved from the EPD New section of the Eukaryotic Promoter Database (EPD) (http://epd.vital-it.ch (accessed on 24 July 2022)) [12]. This resource allows access to the collection of databases of experimentally validated promoters of several model organisms, for which TSS mapping was the result of high-throughput experiments such as CAGE and Oligo-capping, resulting in high precision and high coverage. We used sets of ten animal promoters, vertebrates, invertebrates, and insects, namely *H. sapiens, M. mulatta, M. musculus*, *R. norvegicus*, *C. familiaris*, *G. gallus*, *D. rerio*, *C. elegans*, *D. melanogaster*, and *A. mellifera*; two plant promoters, namely *A. thaliana* and *Z. mays*; two unicellular fungi promoters, namely *S. cerevisae* and *S. pombe*; and protozoan promoters, namely *P. falciparum*. The profiles of the averaged textual, structural, mechanical, and physicochemical properties of 80 bp core promoter sequences (positions from −50 to +30) were constructed.

### 2.1. Comparative Statistical Characteristics of the Nucleotide Sequences in the Core Promoters of Metazoans, Plants, Unicellular Fungi, and Protozoan

First, we compared the percentages of the A, T, G, and C nucleotides in core promoter sequences in different organisms. For simplicity, according to IUPAC nomenclature, we will use the terms W (for nucleotides A and T) and S (for nucleotides G and C). Frequencies of mononucleotides occurrence at each position along the coding strand are shown in Figure 1A–D.

The frequencies of occurrence of dinucleotides in the core promoter sequences of all fifteen species are shown in Appendix A. The frequencies of occurrence of tetranucleotides TATA and AAAA in the core promoter sequences of all fifteen species are shown in Appendix A.

The logo-representation of the promoter sequences with an information content of 1.0 bits is shown in Figure 2, while that with an information content of 0.4 bits is shown in Appendix A. We present two options for scaling the logo image to best reveal the features of different fragments of core promoters because the frequencies of occurrence of nucleotides differ sharply in different regions. Logos were made at http://weblogo.threeplusone.com (accessed on 24 July 2022).

For all of the considered mammalian promoters (*H. sapiens*, *M. mulatta, M. musculus*, *R. norvegicus, C. familiaris*), as well as for promoters of *G. gallus*, the percentage of S exceeds that of W in all positions, for the exception of the TATA element, where the percentages of W are almost equal to that of S (Figure 1A). On the other hand, the promoters of *A. mellifera*, as well as promoters of *A. thaliana,* unicellular fungi *S. cerevisae* and *S. pombe*, and protozoan *P. falciparum* have the highest percentage of W nucleotides at all positions (Figure 1B–D). The promoters of another insect, *D. melanogaster*, as well as promoters of *C. elegans* and *D. rerio*, are composed of a roughly equal amount of W and S nucleotides, while the TATA element is also enriched by W nucleotides. The promoters of another plant, *Z. mays,* have a noticeable asymmetry in the distribution of G and C nucleotides between the coding and non-coding strands. In the coding strand, the content of cytidines is ~15% higher than the content of guanines. This determines both the highest frequency of occurrence of the CC dinucleotide before and after TSS (Appendix A) and the extremely low frequencies of the occurrence of TATA and AAAA tetranucleotides (Appendix A). Another distinguishing feature of *Z. mays* promoters is the presence of a well-defined motif in the vicinity of the +25 position in Figure 1C, Figure 2 and Appendix A. This was also noted earlier [13], where the cap analysis of the gene expression (CAGE) was used to identify genome-wide TSSs in root and stem tissues of two maize (Z. *mays*) inbred lines (B73 and Mo17). The authors hypothesized that the region around +25 harbors an element other than the GC-rich motif that correlates with the presence of TATA consensus. The profiles of all of the species except for *S. cerevisiae* have two regions where the frequencies of dinucleotides occurrence deviate from the mean values (Appendix A). These two regions are located at the TATA-box position and at the region around TSS.

Logo representation (Figure 2) provides detailed information about the characteristic features of the TATA elements and the INR elements in the promoters of each organism. In the position of the TATA element of all mammals, as well as of *G. gallus*, all four nucleotides (G, C, A, and T) occur with equal frequency. In other considered organisms (with the exception of *S. cerevisae*), the frequency of nucleotides A and T in the TATA element are higher than that of G and C. However, the degree of the excess differs quite noticeably between organisms in this group. In both insects (*D. melanogaster* and *A. mellifera)*, it is minimal, and it is most pronounced in *D. rerio*, *A. thaliana*, and *S. pombe*. The logo image of *P. falciparum* differs sharply from all other organisms since the frequencies of the occurrence of the A and T nucleotides are significantly higher.

The occurrences of various octanucleotides in the position of the TATA element of all organisms under consideration are shown in Table 1, while Appendix A also includes the absolute number of each of the octanucleotides in that position for every organism and also presents the frequencies of the occurrence of various octanucleotides in the positions −10–−3 in the promoters of *S. cerevisae*.

We have chosen the TATA-box position in the promoters of each organism based on the positions of the minimum in the profiles of the physical parameter “Stacking energy” and of the maximum in the profiles of the physical parameter “Mobility to bend towards major groove”, which we present in Figure 3, Figure 4, Figure 5 and Figure 6 (lines a,f). A perceptible shift in the position of the TATA box for *A. thaliana* promoters coincides with the data obtained earlier [14].

From Table 1, one can see that the frequencies of occurrence of different octanucleotides presenting the TATA box are rather close. The leading position in this list for all of the analyzed mammalians, as well as in *D. melanogaster* and *C. elegans*, is occupied by the TATAAAAG sequence; however, other octanucleotides occur with a very close frequency. So, the term consensus only conditionally reflects the real situation. Analysis of the TBP-TATA box minor groove interface based on the crystallographic results of their complex structures obtained with refinement better than 2 Å [15] have shown that van der Waals interactions between nonpolar atoms and between nonpolar and polar atoms are factors for complex formation. Moreover, from the kinetic probing, it was found that TBP has less than a 10^3^-fold preference for binding TATAWAAR sequence compared to binding of nonspecific yeast genomic DNA [16]. These results allow us to suggest that hydrogen bonding does not play any role in TBP–TATA box complex formation. Therefore, **those octanucleotides that are selected on the basis of low energy costs for bending towards a wide groove can be TATA elements.**

In contrast, the INR element of all of the organisms is highly selective for the nucleotide sequence. The details can be seen in the logo representation (Figure 2 and Appendix A) and Table 2, Table 3 and Table 4.

From Table 2, one can see that all of the organisms show a preference for PyPu in positions −1 and +1. However, it should be noted that the occurrence of PuPu and PyPy in mammals, *G. gallus*, *D. rerio*, as well as in the plant *Z. mays* is also high enough, noticeably higher than in both insects (*D. melanogaster* and *A. mellifera*), in the plant *A. thaliana,* in the invertebrate *C. elegans*, and in unicellular organisms (*S. cerevisae*, *S. pombe*, and *P. falciparum)*. We find it interesting that the promoters of pure lines of plant *Z. mays* are somewhat different from the promoters of wild-type *A. thaliana*.

All of the organisms, with the exception of *S. cerevisae*, *S. pombe*, and *P. falciparum*, display CA in this position as preferable. In both unicellular fungi (*S. cerevisae* and *S. pombe*), dinucleotides CA and TA are presented in equal amounts in the positions of −1 and +1, while *P. falciparum*, as expected, prefers dinucleotide TA.

What properties of PyPu dinucleotides and especially CA dinucleotide determine their preference in position (−1, +1)? This position is responsible for the double helix divergence, so the dinucleotide step that it occupies must have unique properties. It is known that the deformability of dinucleotides decreases in the order of PyPu > PuPu > PuPy. It was shown that with the help of a spin probe while studying the effects of nucleotide sequence on DNA duplex dynamics [17]. The special mobility of PyPu steps is explained by the greater intensity of the S↔N dynamics in furanose cycles in 5′-terminal pyrimidines compared to 5′-terminal purines, and after 5′Cyt, it reaches its maximum [18]. The advantage of the CpA step over CpG in positions −1 and +1 can be explained by the presence of only two hydrogen bonds, which must be broken at the initial stage of chain divergence. This explanation is confirmed by reactivity with the conformation-sensitive reagent chloroacetaldehyde, which reacts with unpaired adenines and cytosines. This reactivity was confined strictly to adenosine in the d(CA/TG) repeat [19]. In this regard, it is interesting to note that during the formation of nucleosomes, two conformational flexible pyrimidine–purine steps can act as strong positioning signals. These are the pyrimidine–purine step CA/TG, which is unique to the 10 possible dinucleotides and is located preferentially at both inward- and outward-facing minor grooves but not in between, and TA, which is located at inward-facing minor grooves [20].

The occurrence of tetranucleotides in positions −2 and +2, specific for each of the 15 species, is shown in Table 4. It can be assumed that the greater the percentage of less deformable dinucleotides (PuPu or PuPy) in the TSS position of promoter samples of a particular organism, the more variable the strength of different promoters in this organism will be.

### 2.2. Physical and Structural Anisotropy of the Naked DNA in the Core Promoters

The heterogeneity of any DNA fragment is the result of the variation of the physical and structural characteristics of individual base-pair steps. Bending anisotropy, for example, is sequence-dependent and, to a first approximation, reflects both the geometry and stability of the individual base steps [20]. We have built profiles of the base step characteristics for the sets of the core promoters of all 15 organisms using indexes of numerical parameterization for the ten double-stranded duplexes, which are collected in the database DiProDB http://diprodb.fli-leibniz.de (accessed on 24 July 2022) [21]. Among the parameters of a large number of different properties of the ten double-stranded duplexes, which are held in the database, we chose six parameters most suitable for evaluating the anisotropy of nucleotide sequences for DNA axis bending. They are the stacking energy, Roll and Slide, the stiffness of the structure to Roll alteration and to Slide alteration, as well as the stiffness of the structure to bend towards the major groove, which includes alteration to all of the base-pair steps parameters. The database contains several versions of the parameters of the same name, and earlier [10], we verified that the profiles built from different versions of the parameters are in qualitative agreement with each other. Profiles of physical and structural parameters are presented in Figure 3a–f, Figure 4a–f, Figure 5a–f, Figure 6a–f and Figure 7a–f.

We present the profiles of the variations in the stacking energy (Figure 3a, Figure 4a, Figure 5a, Figure 6a and Figure 7a) and the base-pair step parameters of Roll and Slide (Figure 3b–d, Figure 4b–d, Figure 5b–d, Figure 6b–d and Figure 7b–d) in the parametrization of Perez et al. [22], the profiles of stiffness variation in the DNA double helix to Roll and Slide changes (Figure 3c–e, Figure 4c–e, Figure 5c–e, Figure 6c–e and Figure 7c–e) in the parametrization of Goni et al. [23]. These five parameters describe DNA at the base-pair step resolution. To evaluate the stiffness of the structure to bend towards the major groove, we used the parametrization of Gartenberg and Crothers [24]. Their parameter “Mobility to bend towards major groove” was resolved for all 16 dinucleotides and related to each of the complementary strands. In Figure 3f, Figure 4f, Figure 5f, Figure 6f and Figure 7f, this characteristic is presented for the upper strand (the strand complementary to the template). While Figure 3, Figure 4, Figure 5 and Figure 6 present the profiles of the characteristics of the core promoters for all of the 15 organisms. Figure 7 presents the profiles of the same characteristics of two non-promoter regions in *H. sapiens* genomic sequences: the regions (−500–−420) and(−300–−220), and the profiles of the 80 bp set of 30,000 computer-simulated random nucleotide sequences. They are presented along with the profiles of the *H. sapiens* core promoters.

***Stacking energy*** is a part of the enthalpy of DNA formation and defines its stabilizing forces. Its value in the core promoter sequences of all of the mammalians and *G. gallus* is about −16.5 ± 0.2 Kkal/mol (Figure 3a) and Figure 4a), while in invertebrates and unicellular fungi, the stacking energy is somewhat lower (Figure 4a). In plants, the value of the staking energy is intermediate (Figure 5a). The lowest level of stacking energy is in the promoter sequences of *P. falciparum* (Figure 6a). It can be assumed that in this Protozoa, this is due to the compensation of low DNA stability in the absence of a third hydrogen bond in AT-rich sequences. A shallow global minimum on the stacking energy profiles in the region around −28 bp–−34 bp relative to TSS (depending on the organism) is present in the profiles of all organisms, with the exception of *C. elegans*, *A. melifera*, and *S. cerevisiae*. In *P. falciparum*, its depth is the smallest. The good base stacking in the TATA box region is the property of the majority of the specially selected sequences of naked DNA. This is confirmed by the absence of local minima in the stacking energy profiles of the non-promoter regions, as well as in the profiles of the random sequences (Figure 7a). It is interesting that the average level of the stacking energy in the non-promoter regions of the human genome is practically the same as in the promoter regions, while in the set of random sequences, it is somewhat lower. We assume that this is due to the percentage of the AT pairs in the sequences: in the human genome, the percentage of AT pairs is less than the percentage of GC, while in the random sequences, the AT and GC content is approximately the same.

***Base-pair step parameter Roll*** defines an angle between the average planes of two neighboring base pairs. The positive value of this angle corresponds to its opening towards the minor groove. Among the three rotational parameters (helical Twist, Roll, and Tilt), Roll is the most important for understanding the bending of DNA [23,25].

***Base-pair step parameter Slide*** defines the mutual displacement of the neighboring base pairs in the direction perpendicular to the minor and major grooves. The Positive Slide values are a distinguishing feature of B-DNA, while in the A-form of DNA, the values of the Slide are always negative. Thus, the sign of the Slide is an important indicator that allows us to discriminate between the B- and A-DNA forms [26,27].

The values of these two parameters show that the structure of the naked DNA double helix in the promoter regions of mammals, invertebrates, plants, and unicellular fungi (with the exception of their INR element) belongs to the B family. In fact, the structural parameters of Roll and Slide in the core promoter regions of the mammals and *G. gallus* vary between 1.35–1.7° and 0.25–0.48 Å, respectively. In the core promoters of *A. thaliana,* the values of Roll and Slide are somewhat lower than in mammals, especially Slide (~0.2 Ǻ), but in the promoters of another plant, *Z. may*, the values of these parameters are as in mammals. In the promoter sequences of unicellular fungi, the values of Roll and Slide are also close to mammals. The exception is *P. falciparum*. In this Protozoa, double-stranded DNA, at least in the core promoter region, which we have analyzed, may represent the intermediate form with a negative value of Slide, which corresponds to some structure on the B↔A transition path [26,28].

Our profiles show that the values of Roll and Slide, as well as their stiffness in the TATA-box position of all the species (except for *S. cerevisiae*), differ from the average level. The extent of the difference depends on the organism. It is most pronounced in plants, *S. pombe*, and most mammals. The invertebrates present maximum diversity in the TATA-box position. For example, the profiles of Slide and its stiffness of *C. elegans* do not have peculiarities in the TATA-box position, but Roll and its stiffness have. It is important to note that while the values of both structural parameters — Roll and Slide — are somewhat less than the average level, the rigidity of the Roll drops noticeably, while the rigidity of the Slide either remains at an average level or increases. Hence, **it can be concluded that binding to TBP is accompanied by an increase in the opening of the angle between adjacent base pairs towards minor grooves**. This is what happens when the helical axis is bent towards the major grooves. The profiles of the parameter “Mobility to bend towards major groove” in the core promoters of all the organisms (Figure 3f, Figure 4f, Figure 5f and Figure 6f), with the exception of *S. cerevisiae*, clearly reflect this predisposition for octanucleotides in the TATA-box regions. It should be noted that in the core-promoter sequences of *A. melifera*, the increase in the values of the “Mobility to bend towards the major groove” parameter is noticeably less than in other invertebrates. Moreover, in the profiles of *S. cerevisiae*, the maximum falls on the position of −8 bp.

### 2.3. Variations of Ultrasonic Cleavage and DNase I Cleavage Intensities in Core Promoter Sequences

The intensities of the sequence-specific ultrasonic cleavage of the double-stranded DNA provide information on the intensity of the intramolecular conformational movements in every strand [18,29,30], and the DNase I enzymatic cleavage of the double-stranded DNA provide information on the width of their grooves [31,32,33,34]. Therefore, the variation in the local structure in the DNA double helix can also be assessed using the data of these independent new methods.

The relative intensities of the cleavage of the central phosphodiester bond in the 16 dinucleotides and 256 tetranucleotides were determined by multivariate statistical analysis [18]. The experimental details are also given in [29,30]. It was shown that the cleavage rates for all pairs of complementary dinucleotides are significantly different, and the sequence-dependent ultrasonic cleavage rates are consistent with the intensity of N↔S interconversion at the 5′-sugar ring [18]. Therefore, cleavage rates may be useful for characterizing the functional regions of the genome as a measure of local conformational dynamics. We use several indexes for the description of the intensity of ultrasonic cleavage [10]: **R** is the relative cleavage intensities of the central position of each of the 16 dinucleotides; **T** is the relative cleavage intensities of the central position of each of the 256 tetranucleotides; **S** is the combination of indexes **R** and **T** (**S = T − R**). The **S** index provides information on the effect of the nearest context on the intensity of ultrasonic cleavage in the dinucleotide, i.e., if **S** < 0, the first and the fourth nucleotides of a tetranucleotide bring down the intensity of the cleavage in the central step; otherwise they increase it.

The cutting rates of bovine pancreatic deoxyribonuclease I (DNase I) vary along a given DNA sequence, indicating that the enzyme recognizes sequence-dependent structural changes in the DNA double-helix. The high-resolution crystal structures of the two DNase I-DNA complexes showed that the enzyme binds tightly in the minor groove and to the sugar–phosphate backbones of both strands, thereby inducing widening in the minor groove and bending towards the major groove [31,32]. The context near the dinucleotide step strongly affects its cleavage efficiency. These can be rationalized by the fact that six base pairs are in contact with the enzyme. The intrinsic rate of the cleavage by DNase I closely tracks the width of the minor groove [33]. We have used the intensity indices of DNase I cleavage at the hexanucleotide level (D), which were obtained in [34].

Figure 8, Figure 9, Figure 10 and Figure 11 show the profiles of the ultrasonic cleavage and DNase I cleavage in *H. sapiens*, *D. melanogaster*, *Z. mays*, and *P. falciparum*, while Appendix A show the profiles of the ultrasonic cleavage and DNase I cleavage of *M. mulatta*, *M. musculus*, *R. norvegicus*, *C. familiaris*, *G. gallus*, *D. rerio*, *C. elegans*, *A. melifera*, *A. thaliana*, *S. cerevisiae*, and *S. pombe*, respectively.

The profiles of the ultrasonic indexes **R**, **T**, and **S** and the DNase I cleavage index **D** are depicted in blue for the upper strand and in red for the lower (template) strand.

The lowest value of the ultrasonic cleavage for the *H. sapiens* core promoters was detected in the region from −32 to −24 bp relative to TSS (Figure 8, indexes **R** and **T**). The same region of the promoter has the highest DNase I cleavage (Figure 8, index **D**). This indicates a decrease in the conformational motion in this region and minor groove widening. The minimum ultrasonic cleavage of the upper (coding strand) falls at position −26, but in the lower (template) strand, at position −29. This means that there is some shift in the intensity of the conformational movement in the complementary strands. The profiles of the differences in the **S**-indexes between the strands revealed periodic alteration to the conformational motion intensity in the complementary strands until the position of −3 bp. The observed behavior of the core promoter fragment structure is in good agreement with the results of the MD calculations in [35], which confirmed an important role of the indirect readout mechanism in TATA-box recognition, and revealed regular oscillations between several alternate structures in the process of TBP binding.

All of the profiles lose their smoothness around TSS.

The profiles of the ultrasonic cleavage and DNase I cleavage of all of the other mammalian (*M. mulatta, M. musculus*, *R. norvegicus*, and *C. familiaris*), as well as of *G. gallus*, *D. rerio*, *D. melanogaster, A. thaliana, Z. mays*, and *S. pombe* are presented in Appendix A, respectively.

It is significant that the cleavage intensities of the TATA element, as well as that of Inr, have singular properties in the profiles of all but one species. Ultrasonic cleavage diminished in the TATA element, while DNase I cleavage enhanced. The exception is the TATA region in the core promoters of *S. cerevisae.* Both methods show a messy pattern of cleavage around the TSS in all species.

## 3. Discussion

Previously, we found a special structural organization in the nucleotide sequences of double-stranded DNA of minimal core promoters of POL II in metazoans and *Schizosaccharomyces pombe*. They have singular mechanical and structural properties at the positions of the TATA-box and around TSS [10].

This work was undertaken due to the fact that new data appeared that significantly expanded the range of organisms available for analysis, as well as the significant increase in the number of promoter nucleotide sequences available. As a result, the characteristics of the mechanical and structural properties of the core promoters of POL II in the fifteen organisms from different steps of the evolutionary ladder were obtained. These are the ten representatives of the animal kingdom—mammals, vertebrates, and invertebrates— namely, *H. sapiens*, *M. mulatta*, *M. musculus*, *R. norvegicus*, *C. familiaris*, *G. gallus*, *D. rerio*, *C. elegans*, *D. melanogaster*, and *A. mellifera*; two representatives of the plant kingdom (*A. thaliana* and *Z. mays*), two representatives of the kingdom of unicellular fungi (*S. cerevisiae* and *S. pombe*), and a representative of Protozoa (*P. falciparum*). The AT and GC contents of the genomes of these organisms are different. Some of them have a GC-rich genome, while the genomes of the others contain nearly equivalent amounts of AT and GC, or a slight excess of AT, while 80% of the *P. falciparum* genomic sequences consist of AT. The aim of the present work was to assess the generality of the characteristics of the core promoters obtained earlier based on the analysis of a much wider range of organisms that differ significantly in evolutionary development and the percentage of AT pairs in the genomic DNA.

As a result, here we have shown that the core promoters of POL II in organisms representing the kingdoms of animals, plants, fungi, and protozoa have a special structural organization. The fragments of 80 bp (positions from −50 to +30), regardless of the AT content in the genomic DNA, have two singular regions: a hexanucleotide with coordinates −2–+4 (INR) surrounding the transcription start site (TSS) and an octanucleotide separated from TSS at a distance of about 28–35 bp (depending on the organism) located upstream. In the TSS position (−1, +1), the occurrence of the PyPu/PyPu steps is exceptionally high, with a noticeable predominance of the d (CA/TG) dinucleotide. The conformational features of this dinucleotide remarkably favor the formation of an open complex (PIC). The TATA-box region of all but one organism is about 28–35 bp upstream and has unique mechanical and structural properties. Its mobility to bend towards the major groove is increased, and the stacking energy is reduced; the minor groove expands significantly, and the conformational dynamics are reduced. These local properties of the TATA region contribute to its indirect readout by TBP and the subsequent PIC formation.

It is important that the profiles of the control fragments of the same length, taken from the human genome in the vicinity of −300 and −500, as well as from a sample of 30,000 random sequences, do not reveal any structural organization.

However, it should be noted that there is no TATA-element in the position around −28 bp in the promoters of *S. cerevisiae.* However, the structural features that resemble the TATA box are found in the profiles of *S. cerevisiae* at positions −3–−10. We also reveal three organisms (*C. elegans*, *A. melifera*, and *P. falciparum*), where the TATA-element in the position around −28 bp is present, but some of its features are less pronounced. Let us consider in more detail the features of the TATA element in these organisms.

*C. elegans* does not have any peculiarities in the TATA-box position in the profiles of Slide and Slide stiffness, while in the profiles of Roll and Roll stiffness, it has. The magnitude of the maximum in the profile of the parameter “Mobility to bend towards the major groove” is relatively lower than in other organisms, and the profiles of ultrasonic cleavage and DNase I cleavage in the TATA region have no peculiarities until TSS. We suppose that these features are the result of the fact that not TBP but TBP-like factor CeTLF is used to activate Pol II in *C. elegans* [36,37]. Therefore, the PIC assembly machinery may have its own characteristics.

The profiles of the intensity of the ultrasound cleavage and DNase I cleavage of *A. melifera* do not have any features in the area of the TATA element, and the parameter “Mobility to bend towards the major groove” is noticeably less pronounced than in the profiles of the other invertebrates. *A. melifera* is an insect that is characterized by complex social behavior. Its transcription is still studied insufficiently, and there are little data for understanding the details of this process [38].

The extremely high TA content of the *P. falciparum* genomic sequence (about 80%) does not allow the formation of a completely autonomous structure of the Pol II core promoter, which would not require additional control. In *P. falciparum*, both ultrasonic and DNase I cleavage virtually does not change throughout the entire region upstream to TSS. However, in Figure 6f we saw a faintly pronounced wide maximum in the profile of *P. falciparum* “Mobility to bend towards major groove”. It seems that this is a marker for TBP binding, but it is too weak. Apparently, additional mechanisms are needed to realize gene expression and identify the TATA element in the promoter of *P. falciparum*. The role of G-quadruplexes in gene expression is widely discussed [39]. In addition, the presence of G-quadruplex-forming DNA motifs in the *P. falciparum* genome was shown [40]. This is all the more surprising given that 80% of its genome consists of AT pairs. However, it is obvious that the *P. falciparum* genome must contain some additional mechanisms to facilitate the recognition of the TATA element.

Let us try to figure out how much the deviations in the profiles of these three organisms can fundamentally change the idea of an evolutionarily stable structural organization of RNA polymerase II promoters. Despite the absence of some structural features in the region of the TATA element in these three organisms, one of its characteristics is present in all organisms without exception. This characteristic is “Mobility to bend towards the major groove”. It reaches its maximum in the TATA region (Figure 4f), and the presence of the motifs in the logo representations (Figure 2 and Appendix A) of *C. elegans* and *A. melifera* are evident. Thus, *C. elegans*, *A. melifera*, and *P. falciparum* still have a marker of the TATA element. Note that the messy pattern of cleavage around the TSS is present in all organisms.

The only organism whose promoter sequences do not have the structural markers of the TATA element at a position around −28 bp upstream of the TSS is *S. cerevisiae*. However, we registered the maximum in the profiles of the parameter “Mobility to bend towards major groove” at the position of −8 bp. Previously we have already obtained this result when processing a smaller sample of its promoters [10]. The peculiarity of *S. cerevisiae* transcription machinery may be due to the peculiarities of the functioning of Pol II in this organism, which was discovered when compared with *S. pombe* transcription machinery [41]. The differences in the core promoters’ structural organization of two yeasts may be associated with an evolutionary distance between *S. pombe* and *S. cerevisiae*. Really, these organisms diverged in evolution about 500 million years ago [42]. The features of Pol II functioning during transcription in *S. cerevisiae* have recently been studied in detail [43].

## 4. Materials and Methods

We analyzed the sets of promoters of fifteen evolutionarily different organisms that were retrieved from the EPD New section of the Eukaryotic Promoter Database (EPD) (http://epd.vital-it.ch (accessed on 24 July 2022) [12]. We used sets of the animal promoters (29,597 promoters for *H. sapiens*, 9556 promoters for *M. mulatta,* 25,111 promoters for *M. musculus*, 12,569 promoters for *R. norvegicus*, 6126 promoters for *G. gallus,* 7352 promoters for *C. familiaris*, 16,972 promoters for *D. melanogaster*, 6461 promoters for *A. mellifera,* 10,726 promoters for *D. rerio*, 7120 promoters for *C. elegans*); plant promoters (22,702 promoters for *A. thaliana*, 17,059 promoters for *Z. mays*); unicellular fungi promoters (5117 promoters for *S. cerevisae* and 4802 promoters for *S. pombe*); and protozoan promoters (5597 promoters for *P. falciparum)*. We checked that all of these sequences are 80 nucleotides long and strictly defined. The profiles of the averaged textual, structural, mechanical, and the physicochemical properties of 80 bp core promoter sequences (positions from −50 to +30) were constructed.

For analysis of the structural, mechanical, and physicochemical properties of the core promoter sequences, we use indexes of numerical parameterization for the ten double-stranded duplexes, which were collected from the database DiProDB http://diprodb.fli-leibniz.de (accessed on 24 July 2022) [21]. For the profile construction of the variations in the stacking energy and the base-pair step parameters, Roll and Slide, we used the parametrization of Perez et al. [22], for the profile construction of stiffness variation in the DNA double helix to Roll and Slide changes, we used the parametrization of Goni et al. [23], and for the profile construction of stiffness of the structure to bend towards major groove we evaluated using the parametrization of Gartenberg and Crothers [24].

### Profiles Construction

The X-axes of the profiles define the position relative to the TSS, which was denoted as +1 bp, while negative and positive numbers denote the upstream and downstream regions. The Y-axes present the mean value of a chosen characteristic from the corresponding databases. For textual characteristics, defined at the mononucleotide level, for every 80 positions on the X-axis (numbered: −50, −49, … −1, +1, +2, … +30), the amounts of each type of nucleotides (A, C, G, T) in all core promoters from a set of chosen species are summed up, and the resulting sum is divided by the number of promoters. For the physical or structural characteristics defined at the base-pair step level, or for the ultrasound cleavage rates at the dinucleotide level, for every 79 positions on the X-axis (numbered: −49, −48, … −1, +1, +2, … +30), the values of these characteristics are summed up (for dinucleotides at the positions [(−50, −49); (−49, −48); … (−1, +1); … (+29, +30)], taken from DiProDB (physical and structural characteristics) or from the work [18] (ultrasound cleavage rates at the dinucleotide level) and the resulting sum is divided by the number of promoters. For ultrasound cleavage rates at the tetranucleotide level, for every 77 positions on the X-axis (numbered: −48, −47, … −1, +1, +2,… +29), the values of these characteristics for tetranucleotides are summed up (for tetranucleotides at the positions (−50, −49, −48, −47); (−49, −48, −47, −46); …(−2, −1, +1, +2); …(+27, +28, +29, +30)), taken from the Appendix A to the work [18] and the resulting sum is divided by the number of promoters. For the DNAse cleavage rates at hexanucleotide level, for every 75 positions on the X-axis (numbered: −47, −46, …. −1, +1, +2, … +77, +78), the values of these characteristics are summed up (for hexanucleotides at the positions (−50, −49, −48, −47, −46, −45); ( −49, −48, −47, −46, −45, −44); … (−3, −2,−1, +1, +2, +3); …( +25, +26, +27, +28, +29, +30), taken from Appendix A to the work [34]), and the resulting sum is divided by the number of promoters.

We have written the programs in Python 3.10 for profile construction.

## 5. Conclusions

Eukaryote organisms, regardless of the level of their evolutionary development and the AT content of genomic sequences, have common structural features of the naked DNA in the RNA polymerase II core promoter region. These features are the exceptional heterogeneity and asymmetry of the 3D structure and the inclusion of two singular regions—hexanucleotide (“INR”) around TSS and the octanucleotide (“TATA element”) upstream. The strength of each promoter, to some extent, depends on the nucleotide sequences forming its singular regions. In our opinion, all of the data presented here correspond to the bottom-up approach conception of evolution [44], starting from the physicochemical properties of nucleic and amino acid polymers.

## Figures and Tables

**Figure 1 ijms-23-10873-f001:**
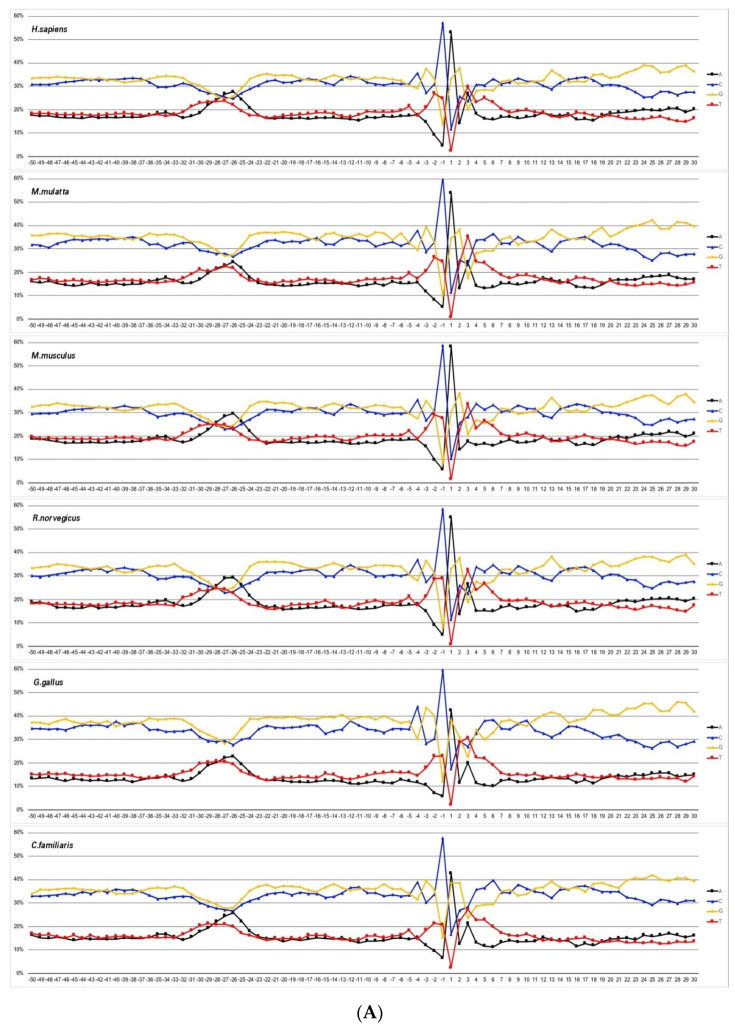
(**A**). Profiles of core promoter sequences as the mononucleotides frequencies of occurrence (in percentages) at each position along the strand, complementary to template for data sets of *H. sapiens*, *M. mulatta, M. musculus*, *R. norvegicus, C. familiaris*, and *G. gallus.* (**B**). Profiles of core promoter sequences as the mononucleotides frequencies of occurrence (in percentages) at each position along the strand, complementary to template for data sets of *D. melanogaster*, *A. mellifera*, *D. rerio*, and *C. elegans.* (**C**). Profiles of core promoter sequences as the mononucleotides frequencies of occurrence (in percentages) at each position along the strand, complementary to template for data sets of *A. thaliana* and *Z. mays*. (**D**). Profiles of core promoter sequences as the mononucleotides frequencies of occurrence (in percentages) at each position along the strand, complementary to template for data sets of *S. cerevisae*, *S. pombe*, and *P. falciparum*.

**Figure 2 ijms-23-10873-f002:**
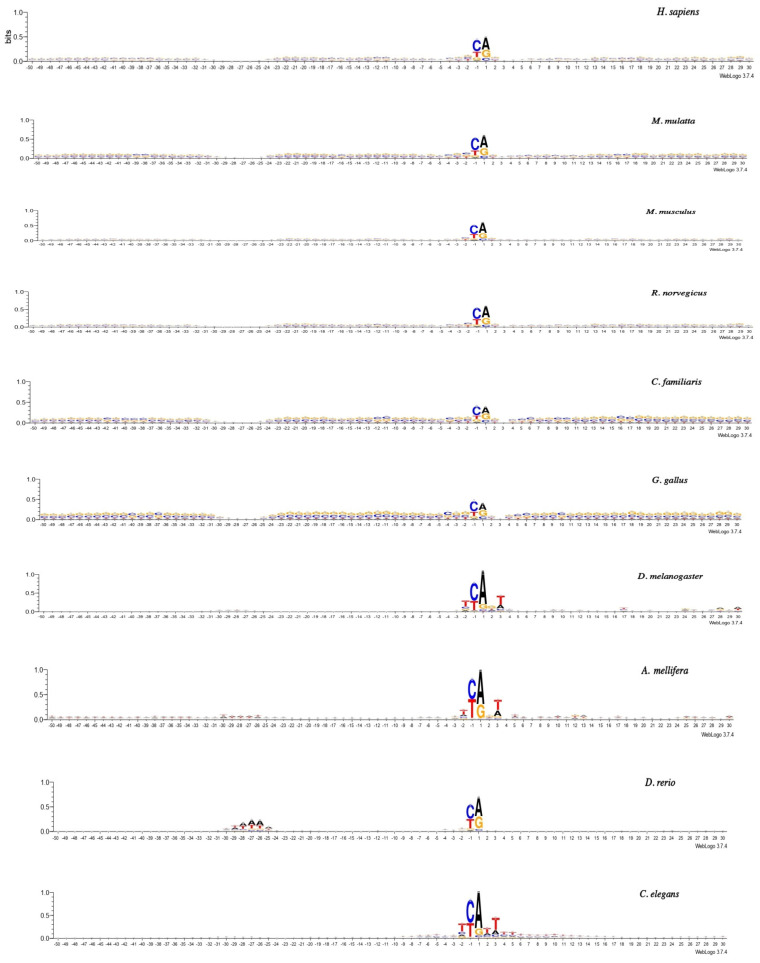
Logo representation with information content 1.0 bits of the promoter sequences of all 15 organisms.

**Figure 3 ijms-23-10873-f003:**
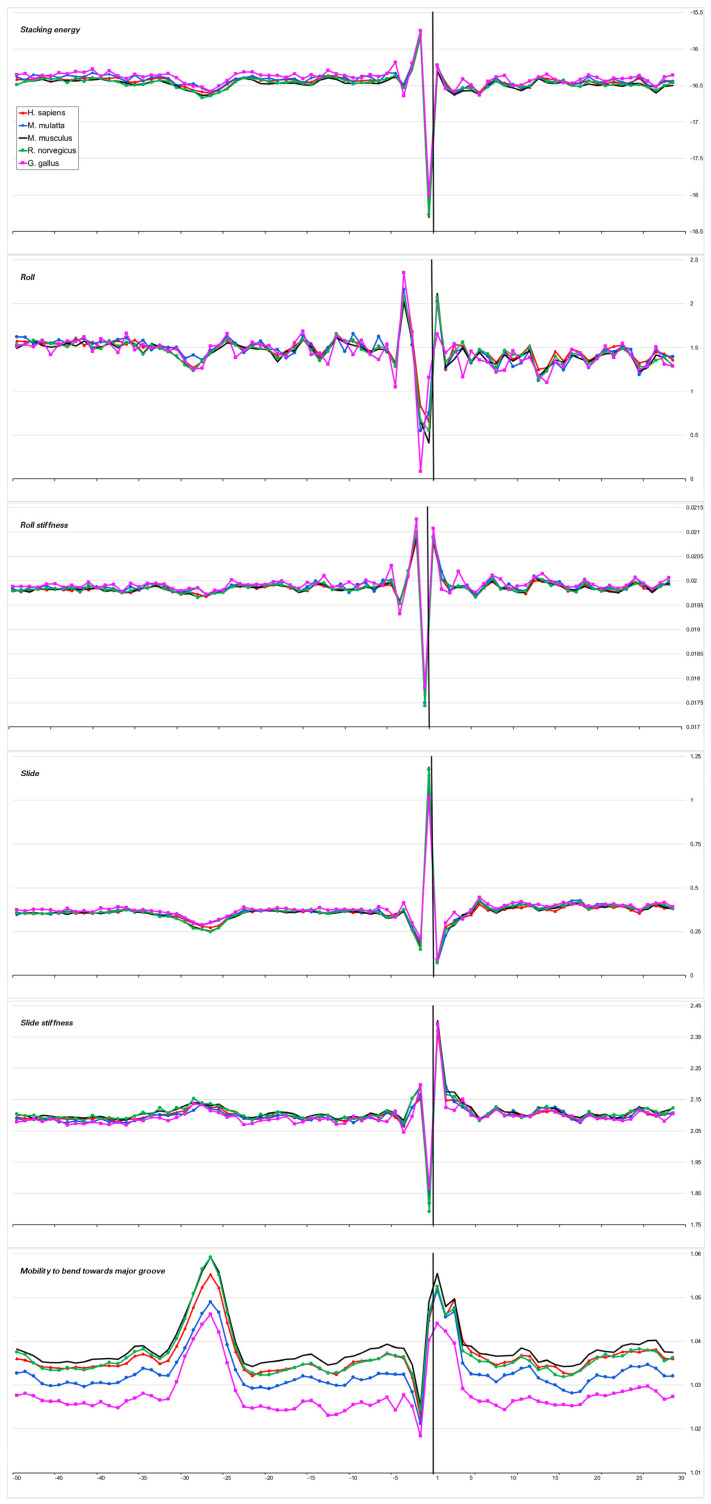
Local variations of the values of physical and structural parameters in core promoter regions of *H. sapiens*, *M. mulatta*, *M. musculus*, *R. norvegicus*, and *G. gallus*. (**a**) Stacking energy (in kcal/mol). (**b**) Roll (in degrees). (**c**) Stiffness of the duplex structure to Roll alteration (in kcal/mol degree). (**d**) Slide (in angstroms). (**e**) Stiffness of the duplex structure to Slide alteration (in kcal/mol angstrom). (**f**) Mobility to bend towards major groove (in mobility units).

**Figure 4 ijms-23-10873-f004:**
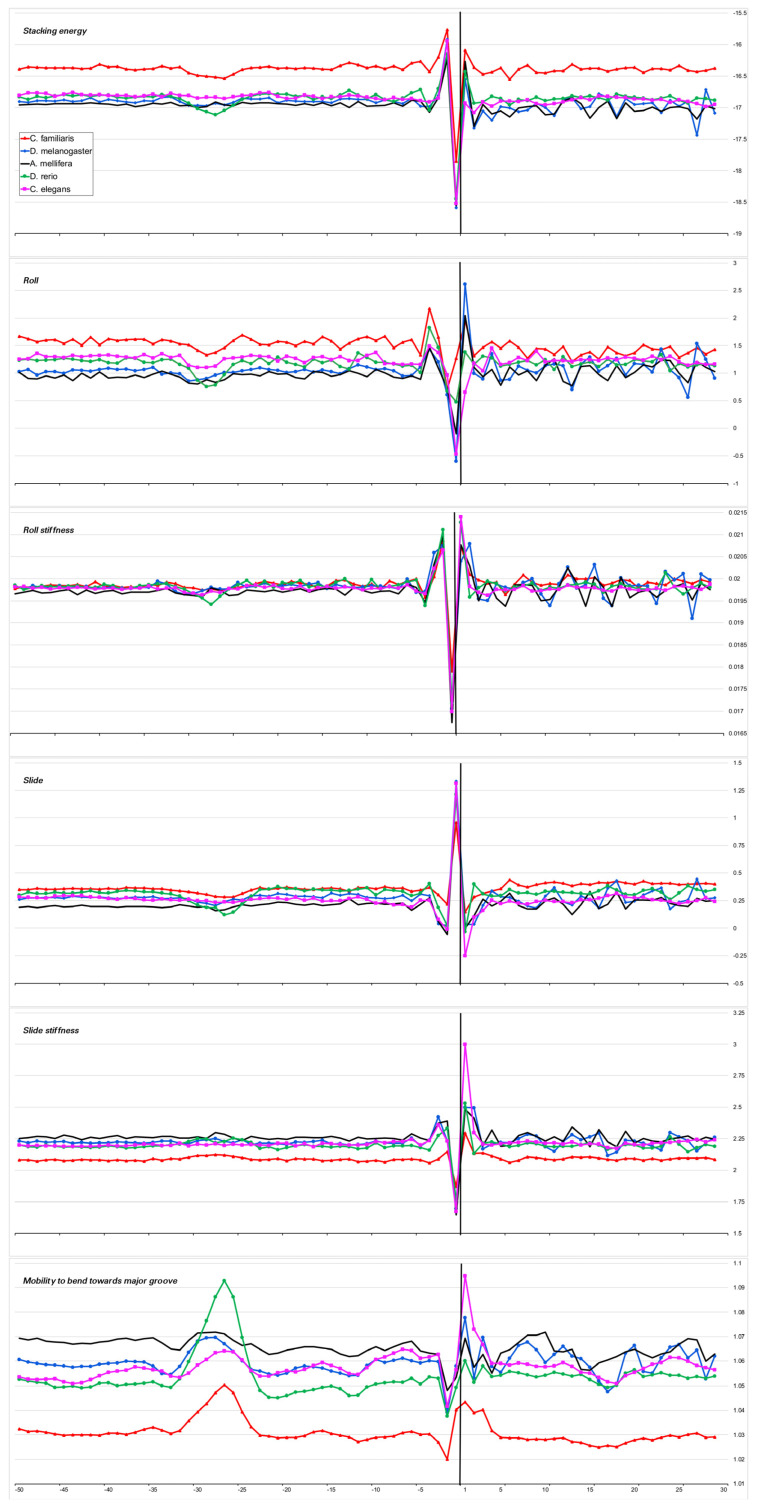
Local variations of the values of physical and structural parameters in core promoter regions of *C. familiaris*, *D. melanogaster*, *A. mellifera*, *D. rerio*, *C. elegans*. (**a**) Stacking energy (in kcal/mol). (**b**) Roll (in degrees). (**c**) Stiffness of the duplex structure to Roll alteration (in kcal/mol degree). (**d**) Slide (in angstroms). (**e**) Stiffness of the duplex structure to Slide alteration (in kcal/mol angstrom). (**f**) Mobility to bend towards major groove (in mobility units).

**Figure 5 ijms-23-10873-f005:**
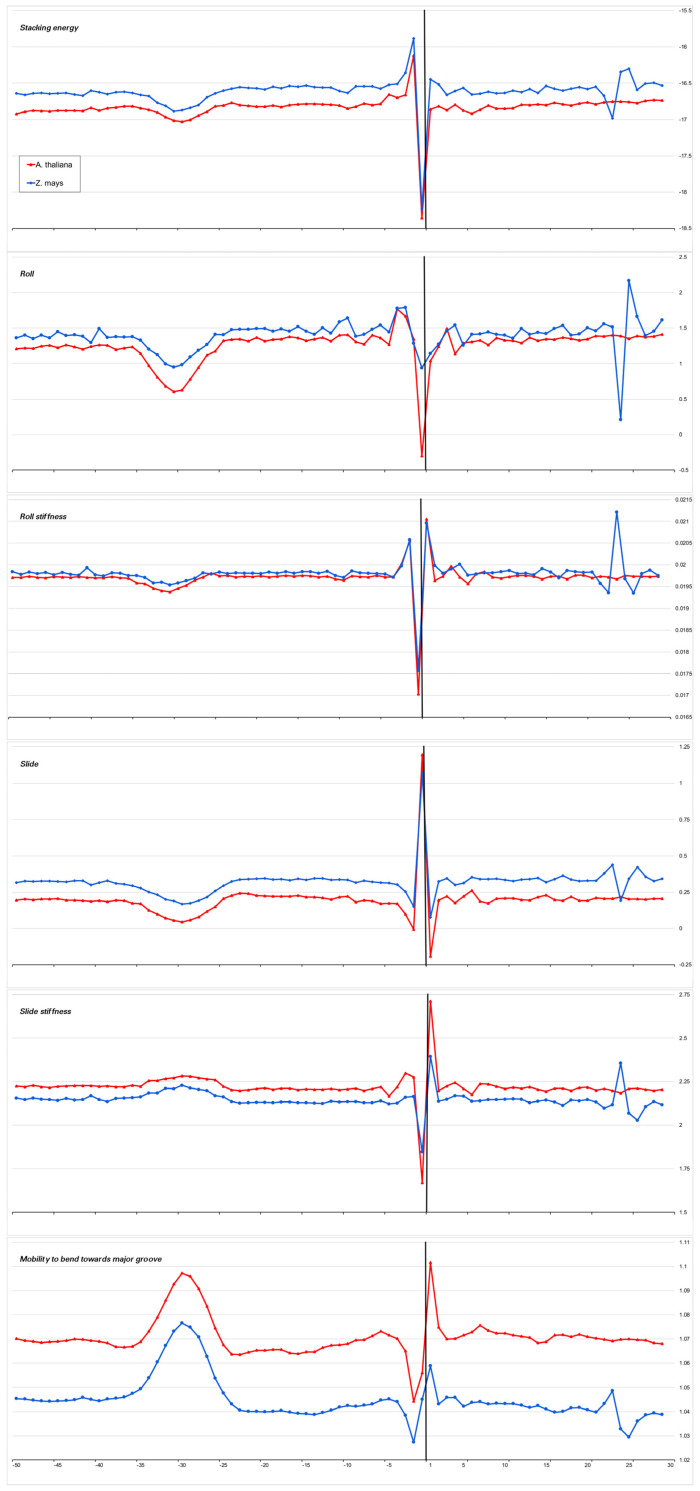
Local variations of the values of physical and structural parameters in core promoter regions of *A. thaliana* and *Z. mays.* (**a**) Stacking energy (in kcal/mol). (**b**) Roll (in degrees). (**c**) Stiffness of the duplex structure to Roll alteration (in kcal/mol degree). (**d**) Slide (in angstroms). (**e**) Stiffness of the duplex structure to Slide alteration (in kcal/mol angstrom). (**f**) Mobility to bend towards major groove (in mobility units).

**Figure 6 ijms-23-10873-f006:**
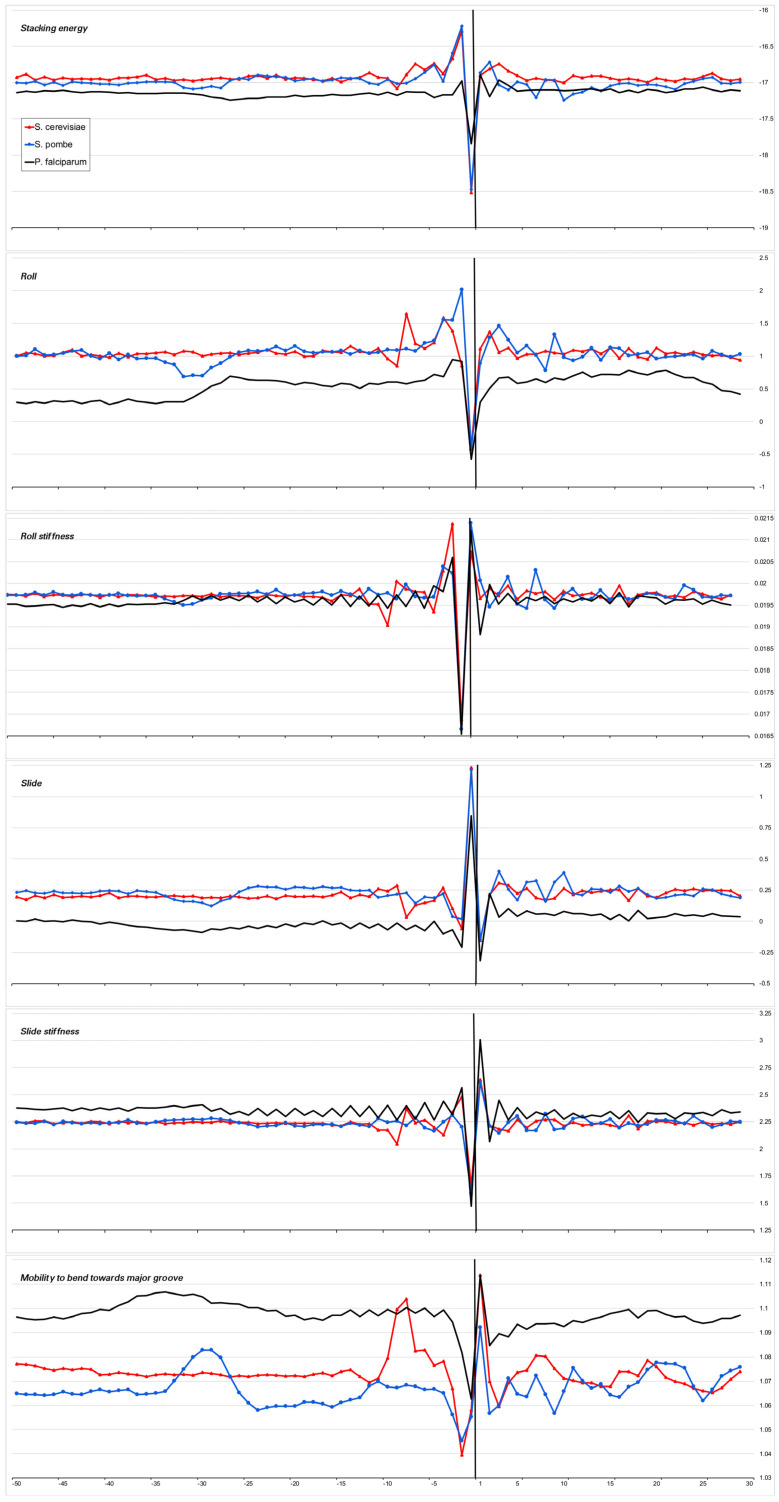
Local variations of the values of physical and structural parameters in core promoter regions of *S. cerevisae*, *S. pombe*, and *P. falciparum.* (**a**) Stacking energy (in kcal/mol). (**b**) Roll (in degrees). (**c**) Stiffness of the duplex structure to Roll alteration (in kcal/mol degree). (**d**) Slide (in angstroms). (**e**) Stiffness of the duplex structure to Slide alteration (in kcal/mol angstrom). (**f**) Mobility to bend towards major groove (in mobility units).

**Figure 7 ijms-23-10873-f007:**
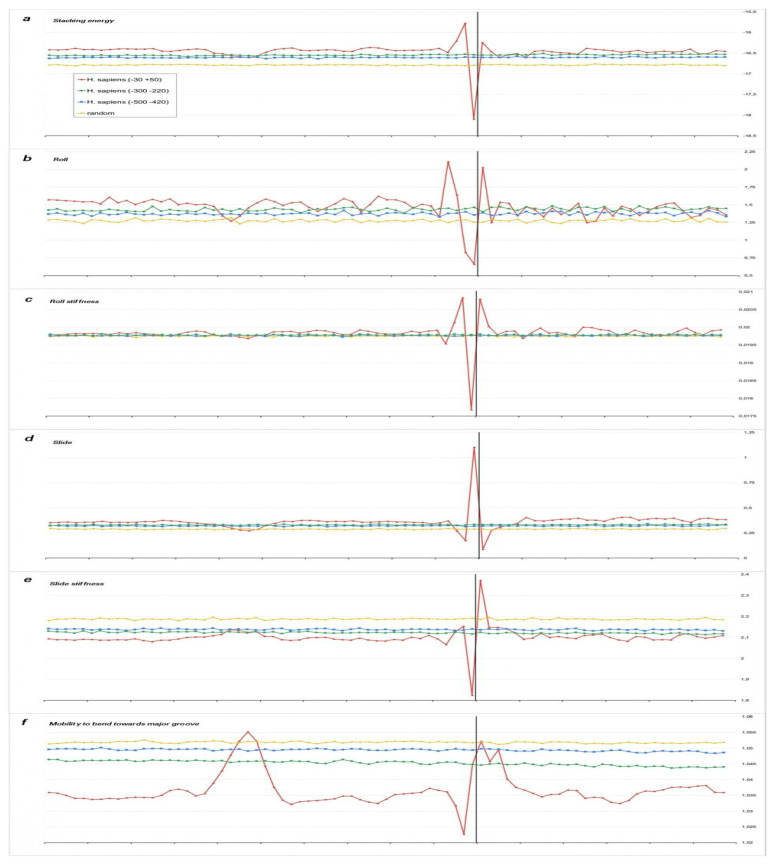
Local variations of the values of physical and structural parameters in two non-promoter regions from *H. sapiens* genomic sequences: the regions (−500–−420) and (−300–−220), and the profiles of the 80 bp set of 30,000 computer simulated random nucleotide sequences along with the profiles of *H. sapiens* core promoters. (**a**) Stacking energy (in kcal/mol). (**b**) Roll (in degrees). (**c**) Stiffness of the duplex structure to Roll alteration (in kcal/mol degree). (**d**) Slide (in angstroms). (**e**) Stiffness of the duplex structure to Slide alteration (in kcal/mol angstrom). (**f**) Mobility to bend towards major groove (in mobility units).

**Figure 8 ijms-23-10873-f008:**
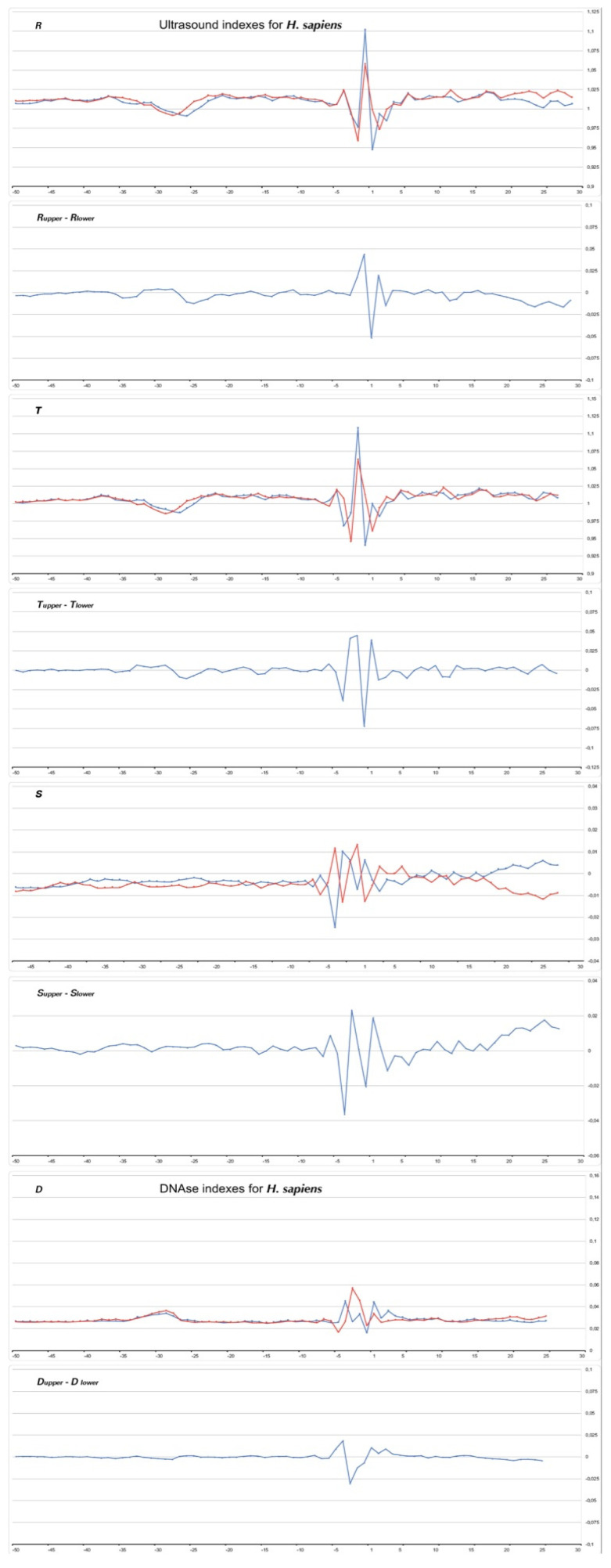
Profiles of ultrasonic cleavage indexes and DNase I cleavage indexes for *H. sapiens* core promoters.

**Figure 9 ijms-23-10873-f009:**
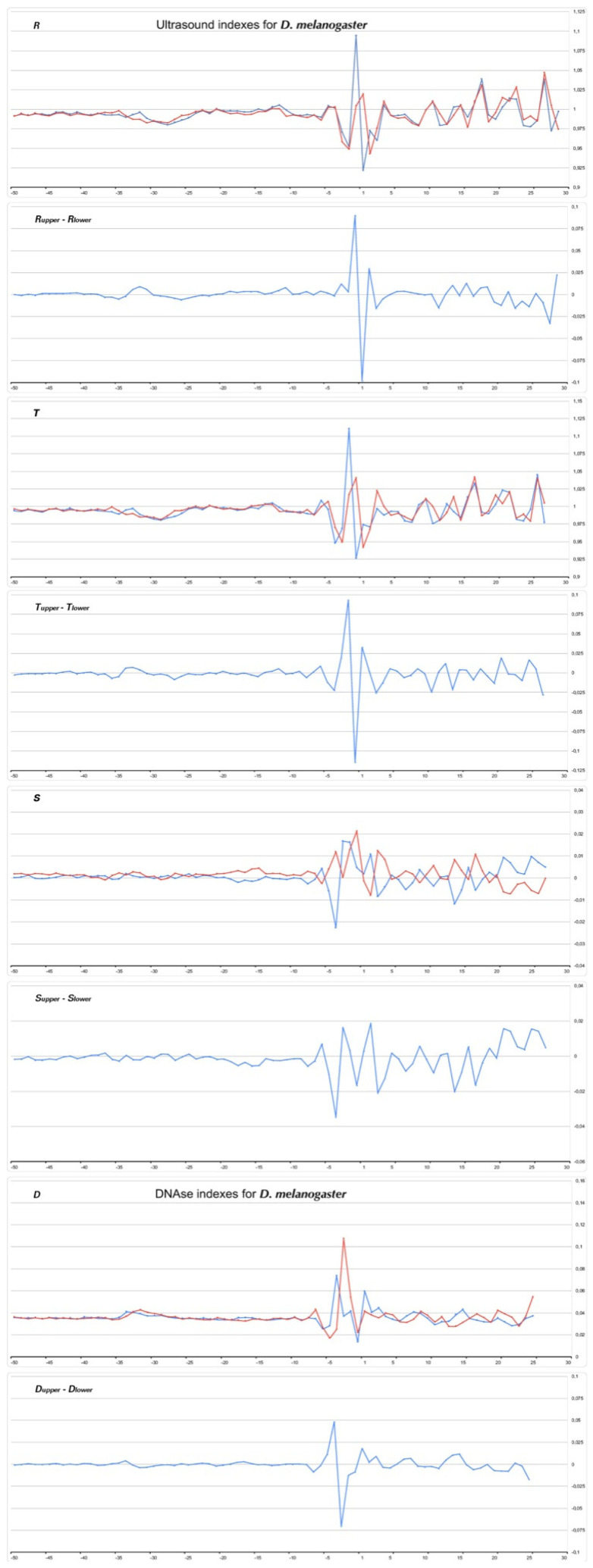
Profiles of ultrasonic cleavage indexes and DNase I cleavage indexes for *D. melanogaster* core promoters.

**Figure 10 ijms-23-10873-f010:**
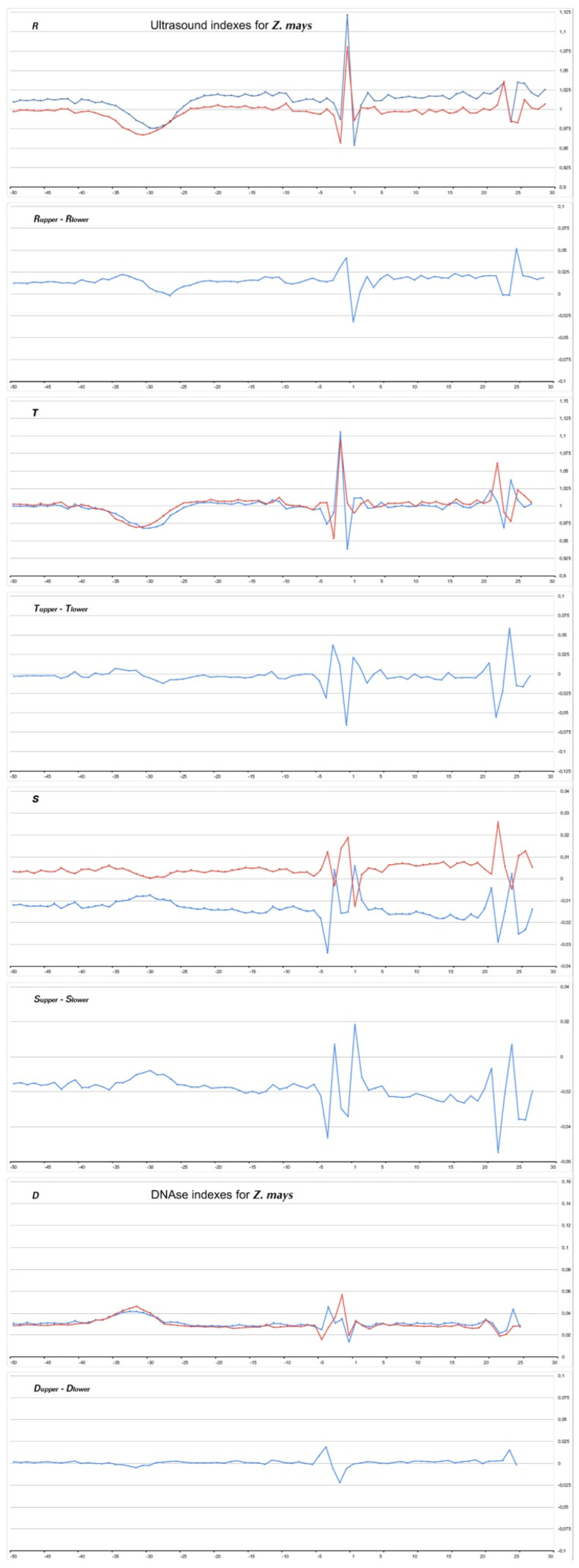
Profiles of ultrasonic cleavage indexes and DNase I cleavage indexes for *Z. mays* core promoters.

**Figure 11 ijms-23-10873-f011:**
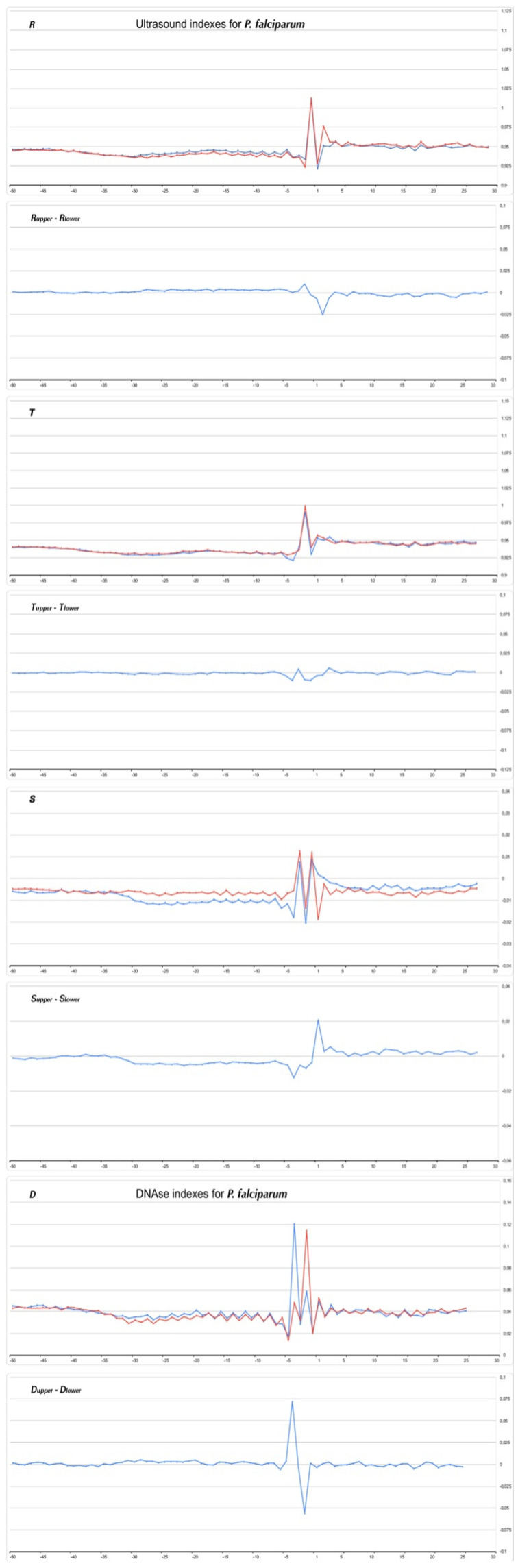
Profiles of ultrasonic cleavage indexes and DNase I cleavage indexes for *P. falciparum* core promoters.

**Table 1 ijms-23-10873-t001:** Frequencies of occurrence of different octanucleotides in TATA-box position of every studied organisms (for the exception of *S. cerevisae*).

	** *H. sapiens* ** **(−30–−23)**		** *M. mulatta* ** **(−31–−24)**		** *M. musculus* ** **(−30–−23)**		** *R. norvegicus* ** **(−30–−23)**		** *C. familiaris* ** **(−30–−23)**		** *G. gallus* ** **(−30–−23)**		** *D. melanogaster* ** **(−31–−24)**		** *A. mellifera* ** **(−32–−25)**	
1	TATAAAAG	0.20%	TATAAAAG	0.14%	TATAAAAG	0.34%	TATAAAAG	0.40%	TATAAAAG	0.20%	GGGGCGGG	0.26%	TATAAAAG	0.84%	TATATATA	0.66%
2	TTTTTTTT	0.12%	GGGGCGGG	0.14%	TTTTTTTT	0.32%	TATAAAGG	0.18%	GGGCGGGG	0.20%	TATATAAG	0.20%	TATAAATA	0.36%	ATATATAT	0.49%
3	ATAAAAGG	0.11%	CGCCGCCG	0.13%	TATATAAG	0.21%	ATAAAAGG	0.16%	TATATAAG	0.16%	TTTTTTTT	0.18%	CTATAAAA	0.35%	TATATATT	0.28%
4	GGGCGGGG	0.11%	GGCGGCGG	0.11%	ATAAAAGG	0.19%	TATAAATA	0.12%	TATAAAAA	0.15%	CCGCCCCG	0.18%	ATAAAAGC	0.34%	CATATATA	0.15%
5	GCCCCGCC	0.10%	CTATAAAG	0.10%	TATAAAGG	0.15%	TATATAAG	0.12%	CCGGAAGT	0.13%	GGCGGGGC	0.18%	GTATAAAA	0.27%	GTATAAAA	0.15%
6	TATATAAG	0.10%	CTATAAAA	0.10%	TATAAATA	0.12%	TATATAAA	0.12%	ATAAAGGC	0.12%	GCGGGGCG	0.18%	CTATATAA	0.26%	TATATAAG	0.15%
7	GGGGCGGG	0.10%	CCCCGCCC	0.10%	ATATAAGG	0.11%	TATAAAGA	0.12%	GCGGCGGC	0.12%	TATAAAAG	0.16%	TATAAAAA	0.24%	TATAAAAG	0.15%
8	TATAAAAA	0.10%	CCGGAAGC	0.10%	ATAAATAG	0.11%	CTATAAAA	0.12%	TATAAATA	0.12%	ATAAAAGC	0.16%	TATATAAG	0.24%	ATATATAA	0.15%
9	TATAAAGG	0.09%	CGGCGGCG	0.09%	ATAAAAGC	0.10%	ATAAAAAG	0.11%	GCCCCGCC	0.12%	GCGGCGGG	0.15%	TATATAAA	0.22%	TATATAAA	0.14%
10	CCCCTCCC	0.08%	TGGGCGGG	0.08%	ATAAAAAG	0.10%	ATAAAAGC	0.10%	CGCCGCCG	0.11%	GATAAAAG	0.15%	ATAAATAG	0.19%	TTATATAT	0.12%
11	CCCCGCCC	0.08%	CCGCCCCG	0.08%	GGGGCGGG	0.10%	TATAAAGC	0.10%	TTTTTTTT	0.11%	TATAAAGG	0.15%	ATATAAAA	0.17%	TATAAATA	0.11%
12	ATATAAAG	0.08%	CTATATAA	0.08%	GCCCCGCC	0.09%	GCCCCGCC	0.10%	GGGGCGGG	0.11%	TATAAAGC	0.15%	GTATATAA	0.14%	CTATATAT	0.11%
13	CTATAAAA	0.08%	CGCCCCGC	0.08%	TATATAAA	0.09%	ATAAAGGC	0.10%	CCCGCCCC	0.11%	TATAAAAA	0.15%	ATAAAAAC	0.13%	TTATATTT	0.11%
14	ATAAAAGC	0.08%	AAAAAAAA	0.08%	TATAAGAG	0.09%	ATATAAAG	0.10%	ATAAAAGG	0.09%	CCCGCCCC	0.13%	TTATAAAA	0.13%	GTATATAT	0.11%
15	CCCGCCCC	0.08%	TTATAAAA	0.07%	ATAAAAGA	0.08%	TATAAGAG	0.10%	ATATAAGG	0.09%	ATAAAAGG	0.13%	ATATAAGC	0.12%	ATGTATAT	0.09%
16	CTATAAAG	0.07%	GCGCCTGC	0.07%	ATATAAAG	0.08%	TAAAAGCC	0.09%	CCCCGCCC	0.09%	TCCCTCCC	0.13%	TATAAAAT	0.12%	TATATGTA	0.09%
17	GAATAAAA	0.06%	GGAGGAGG	0.07%	CTATAAAA	0.08%	GGGGCGGG	0.09%	CCCTCCCC	0.09%	CGGGGCGG	0.13%	ATAAAAGA	0.12%	AGTATATA	0.09%
18	TTAAAAGG	0.06%	GCGGCGCG	0.07%	GATAAAAG	0.08%	AGATAAAA	0.09%	GGCGGCGG	0.09%	CACTTCCG	0.11%	TAAAAGCC	0.12%	ATATAAAT	0.09%
19	TTTAAAAG	0.06%	CATAAAAG	0.07%	TTTAAAAG	0.08%	ATAAATAG	0.09%	GCTTCCGG	0.09%	CGCTTCCG	0.11%	ATAAAAGG	0.12%	ATTATATA	0.09%
20	TATAAGAG	0.06%	GCGGCGGC	0.07%	AATAAAAG	0.07%	TATAAAAA	0.08%	TATAAAGG	0.09%	GCCCCGCC	0.11%	GTATAAAT	0.12%	TAAATATT	0.08%
	** *A. mellifera* ** **(−32–−25)**		** *D. rerio* ** **(−30–−23)**		** *C. elegans* ** **(−31–−24)**		** *A. thaliana* ** **(−34–−27)**		** *Z. mays* ** **(−34–−27)**		** *S. pombe* ** **(−34–−27)**		** *P. falciparum* ** **(−39–−32)**			
1	TATATATA	0.66%	TATAAATA	0.28%	TATAAAAG	0.90%	TATATATA	1.43%	TATATATA	0.60%	TATATATA	0.67%	ATATATAT	5.79%		
2	ATATATAT	0.49%	TTTATTTT	0.22%	GTATAAAA	0.42%	TATAAATA	0.98%	CTATAAAT	0.34%	ATATATAT	0.42%	TATATATA	4.97%		
3	TATATATT	0.28%	TATAAAAG	0.21%	TATAAATA	0.38%	ATATATAT	0.76%	CTATATAA	0.29%	TATATAAA	0.27%	AAAAAAAA	3.73%		
4	CATATATA	0.15%	CTTTTATT	0.20%	CTATAAAA	0.28%	TATATAAA	0.65%	TATAAATA	0.29%	CTATATAA	0.23%	TTTTTTTT	3.59%		
5	GTATAAAA	0.15%	TTTTATTT	0.18%	TATATAAA	0.28%	CTATAAAT	0.60%	ATATATAT	0.29%	CATATATA	0.21%	TATATATT	1.05%		
6	TATATAAG	0.15%	TTTAAAAG	0.17%	TATAAAAA	0.25%	CTATATAA	0.52%	CTATATAT	0.25%	GTATATAT	0.21%	ATATATAA	0.79%		
7	TATAAAAG	0.15%	TATAAAAA	0.15%	ATAAAAGA	0.25%	CTATATAT	0.51%	TATATAAA	0.24%	CTATATAT	0.21%	ATATAATA	0.71%		
8	ATATATAA	0.15%	TATAAAGC	0.15%	GTATATAA	0.24%	ATATATAA	0.46%	CTATAAAA	0.23%	ATATATAA	0.19%	ATATATTT	0.70%		
9	TATATAAA	0.14%	TATAAAAC	0.15%	ATATAAAA	0.21%	TCTATATA	0.41%	CCTATAAA	0.18%	TATATAAG	0.19%	ATTTTTTT	0.66%		
10	TTATATAT	0.12%	ATAAAAGC	0.14%	TATATAAG	0.20%	TCTATAAA	0.39%	GTATATAT	0.15%	ACTATATA	0.17%	TTATATAT	0.63%		
11	TATAAATA	0.11%	TATATAAA	0.14%	TATAAAAT	0.20%	ATATAAAT	0.39%	TCTATATA	0.15%	ATATAAAT	0.17%	TAAATAAA	0.59%		
12	CTATATAT	0.11%	TTATTTTG	0.12%	GTATAAAT	0.20%	TATAAAAA	0.29%	ATATATAC	0.14%	TATAAAAG	0.17%	TTTATTTT	0.57%		
13	TTATATTT	0.11%	TTTAAAAA	0.12%	ATATAAAT	0.15%	TTATAAAT	0.28%	ATATATAA	0.14%	AAACGATG	0.17%	TATATAAT	0.55%		
14	GTATATAT	0.11%	GAGAGAGA	0.11%	AGTATAAA	0.15%	CTATAAAA	0.28%	GCTATAAA	0.14%	GTATAAAT	0.17%	AATAAATA	0.55%		
15	ATGTATAT	0.09%	ACTTTTAT	0.11%	ATAAAAGG	0.14%	TTTATATA	0.27%	ATAAATAG	0.13%	TGAATAAA	0.15%	AATATATA	0.55%		
16	TATATGTA	0.09%	ATAAAAGG	0.11%	GGTATAAA	0.13%	TTATATAT	0.24%	TATAAAAG	0.13%	TGTATATA	0.15%	TATTTTTT	0.55%		
17	AGTATATA	0.09%	ATAAATAC	0.11%	TATAAATT	0.11%	GTATATAT	0.24%	TATATAAG	0.13%	TTAAAAAA	0.12%	TTTTTTTA	0.50%		
18	ATATAAAT	0.09%	TATAAACA	0.11%	ATAAAAAG	0.11%	ATATAAAC	0.23%	TATAAAAA	0.13%	ATATATAG	0.12%	ATATTATA	0.50%		
19	ATTATATA	0.09%	TTTAAATA	0.11%	TATATATA	0.11%	ATAAATAA	0.23%	TATAAAAC	0.12%	TATATATT	0.12%	TTTTATTT	0.50%		
20	TAAATATT	0.08%	TTTAAAAC	0.10%	ATAAATAG	0.10%	TTATATAA	0.23%	ATATAAAC	0.12%	AATATAAA	0.12%	TAAATATA	0.48%		

**Table 2 ijms-23-10873-t002:** The content (percentage) of dinucleotides PyPu, PuPu, PyPy, and PuPy in positions −1, +1.

	PyPu	PuPu	PyPy	PuPy
*H. sapiens*	72.17%	13.83%	9.66%	4.34%
*M. mulatta*	76.49%	11.61%	8.78%	3.11%
*M. musculus*	77.63%	10.73%	8.70%	2.94%
*R. norvegicus*	77.71%	10.19%	9.81%	2.29%
*C. familiaris*	65.34%	15.73%	13.04%	5.89%
*G. gallus*	68.30%	12.39%	13.96%	5.35%
*D. melanogaster*	91.26%	2.87%	3.54%	2.33%
*A. mellifera*	95.40%	2.59%	1.60%	0.40%
*D. rerio*	83.52%	9.97%	5.71%	0.79%
*C. elegans*	90.97%	2.78%	5.67%	0.58%
*A. thaliana*	88.81%	5.97%	3.55%	1.67%
*Z. mays*	75.15%	9.38%	10.29%	5.18%
*S. cerevisiae*	93.59%	2.21%	2.19%	2.01%
*S. pombe*	97.42%	1.10%	1.08%	0.40%
*P. falciparum*	95.09%	1.68%	1.95%	1.29%

**Table 3 ijms-23-10873-t003:** The content (percentage) of each of 16 dinucleotides in positions −1, +1.

	AA	AC	AG	AT	CA	CC	CG	CT	GA	GC	GG	GT	TA	TC	TG	TT
*H. sapiens*	1.65%	0.90%	1.83%	0.25%	38.24%	5.06%	13.41%	0.44%	4.90%	2.04%	5.44%	1.15%	8.18%	3.74%	12.35%	0.41%
*M. mulatto*	2.02%	1.11%	1.80%	0.07%	39.11%	4.88%	16.66%	0.22%	4.43%	1.78%	3.37%	0.16%	8.21%	3.57%	12.51%	0.11%
*M. musculus*	2.25%	1.11%	1.93%	0.30%	42.17%	4.31%	11.82%	0.49%	3.89%	1.18%	2.66%	0.34%	9.96%	3.65%	13.69%	0.25%
*R. norvegicus*	1.76%	0.86%	1.94%	0.11%	39.84%	4.98%	13.55%	0.18%	3.43%	1.13%	3.06%	0.19%	9.96%	4.44%	14.37%	0.20%
*C. familiaris*	2.01%	1.31%	2.86%	0.25%	29.88%	7.36%	19.60%	0.97%	5.32%	3.57%	5.54%	0.76%	5.46%	4.36%	10.39%	0.36%
*G. gallus*	2.27%	1.27%	1.83%	0.36%	31.02%	8.37%	19.57%	0.52%	4.24%	3.09%	4.05%	0.64%	4.93%	4.68%	12.78%	0.38%
*D. melanogaster*	0.62%	0.71%	0.74%	0.74%	57.76%	0.85%	3.34%	0.41%	0.86%	0.58%	0.65%	0.30%	22.95%	1.86%	7.21%	0.43%
*A. mellifera*	0.40%	0.06%	0.77%	0.20%	39.59%	0.59%	10.23%	0.06%	0.57%	0.06%	0.85%	0.08%	28.96%	0.85%	16.63%	0.11%
*D. rerio*	2.03%	0.18%	2.20%	0.03%	36.72%	1.94%	14.04%	0.61%	3.31%	0.45%	2.43%	0.14%	13.82%	2.58%	18.93%	0.59%
*C. elegans*	0.62%	0.15%	0.45%	0.14%	53.76%	1.35%	4.34%	0.74%	1.18%	0.21%	0.53%	0.07%	23.56%	3.01%	9.31%	0.58%
*A. thaliana*	0.96%	0.31%	0.82%	0.27%	43.42%	1.16%	5.97%	0.29%	3.12%	0.48%	1.07%	0.60%	27.08%	1.78%	12.34%	0.31%
*Z. mays*	1.48%	1.45%	2.21%	0.71%	35.00%	4.52%	18.22%	1.74%	3.15%	2.07%	2.54%	0.94%	9.62%	3.20%	12.32%	0.83%
*S. cerevisiae*	0.76%	0.68%	0.65%	0.70%	47.65%	0.66%	6.08%	0.33%	0.43%	0.33%	0.37%	0.29%	30.22%	0.61%	9.64%	0.59%
*S. pombe*	0.21%	0.19%	0.19%	0.00%	36.17%	0.48%	6.40%	0.00%	0.42%	0.13%	0.29%	0.06%	36.30%	0.44%	18.59%	0.15%
*P. falciparum*	1.36%	0.21%	0.14%	1.04%	15.60%	0.18%	1.93%	0.25%	0.16%	0.00%	0.00%	0.04%	60.68%	0.39%	16.89%	1.13%

**Table 4 ijms-23-10873-t004:** The content (percentage) of tetranucleotides in positions −2, +2.

	*H. sapiens*		*M. mulatto*		*M. musculus*		*R. norvegicus*		*C. familiaris*		*G. gallus*		*D. melanogaster*		*A. mellifera*		*D. rerio*		*C. elegans*		*A. thaliana*		*Z. mays*		*S. cerevisiae*		*S. pombe*		*P. falciparum*	
1	CCAG	6.98%	GCAG	7.62%	TCAG	7.39%	TCAG	7.05%	GCAG	6.40%	GCAG	8.32%	TCAG	20.34%	TCAG	12.47%	TCAG	6.74%	TCAT	16.03%	TCAT	7.58%	CCAC	4.64%	ACAA	6.27%	CCAA	6.60%	TTAT	22.78%
2	GCAG	6.49%	CCAG	6.85%	GCAG	7.02%	GCAG	6.79%	CCAG	6.02%	TCAG	3.77%	TTAG	7.30%	TTAG	6.48%	TCAC	4.46%	TTAT	7.46%	TCAA	6.31%	CCAG	3.99%	CCAA	6.06%	TTAC	6.08%	TTAA	12.92%
3	TCAG	5.99%	TCAG	5.64%	CCAG	6.97%	CCAG	6.53%	TCAG	4.14%	GCGC	3.48%	TCAC	5.17%	CCAG	3.67%	GCAG	4.07%	TCAC	6.42%	TCAC	5.55%	CCAA	3.66%	TCAA	5.22%	TTAA	5.60%	ATAT	10.26%
4	TCAC	3.21%	CCAC	3.13%	TCAC	3.62%	TCAC	3.32%	CCGC	3.06%	CCGC	3.28%	TCAT	5.13%	ACAG	3.65%	TCAT	2.93%	CCAT	5.01%	TTAT	3.79%	GCAG	2.96%	GCAA	4.03%	TCAA	5.46%	TTGT	6.72%
5	CCAC	2.92%	TCAC	3.05%	CCAC	3.18%	CCAC	3.13%	GCGG	2.88%	GCAC	3.26%	CCAG	4.61%	TCAT	3.51%	TTGT	2.71%	TCAG	4.27%	CCAA	3.52%	TCAC	2.94%	CCAT	3.89%	CTAC	4.58%	TCAT	4.45%
6	GCAC	2.35%	GCGC	2.44%	GCAC	2.50%	GCGC	2.36%	GCGC	2.88%	CCAG	3.07%	GCAG	4.58%	GCAG	3.22%	ACAG	2.68%	TCAA	4.10%	TTAA	3.45%	TCAG	2.46%	ATAA	3.85%	CTAA	4.41%	TTGA	4.43%
7	ACAG	1.96%	GCAC	2.29%	ACAG	2.33%	GCAC	2.33%	CCAC	2.32%	TCAC	2.63%	ACAG	3.29%	TTGA	3.13%	ACAC	2.40%	TTGT	3.03%	ACAA	3.22%	CCGC	2.41%	GTAA	3.65%	TTGC	3.89%	ATAA	4.40%
8	GCGC	1.91%	CCGC	2.23%	CTGT	1.98%	CTGT	2.04%	CCGG	2.25%	GCGG	2.61%	TTAT	2.75%	TCAC	2.93%	CCAG	2.29%	CTAT	2.77%	CCAT	3.06%	TCGC	2.27%	ATAT	3.21%	TCAC	3.69%	TCAA	2.93%
9	CCGC	1.89%	GCGG	2.17%	GCGC	1.88%	ACAG	2.02%	TCAC	2.15%	TCCT	2.50%	TCAA	2.55%	TTAC	2.82%	TTAC	1.95%	ACAT	2.61%	CTAA	2.89%	CTGC	2.11%	ACAT	3.15%	CCAC	3.56%	TTAG	2.29%
10	GGAG	1.80%	CCAT	2.02%	CCAT	1.79%	TCCT	1.89%	GGAG	2.11%	CCGT	2.14%	TTAA	2.11%	TTAT	2.76%	TTGA	1.83%	CCAC	2.42%	TCAG	2.74%	CCAT	2.11%	CTAЛ	2.89%	TTAG	3.37%	ACAT	2.16%
11	CCAT	1.68%	CTGT	1.99%	TTAG	1.74%	CTGA	1.86%	TCCT	2.00%	CCAC	2.04%	TTGT	2.02%	TTGT	2.70%	GCAC	1.75%	GCAT	2.32%	TTGT	2.47%	GCAC	2.09%	TTAA	2.81%	TTGA	3.31%	TTAC	1.98%
12	TCAT	1.67%	ACAG	1.84%	CTGA	1.59%	CTAG	1.82%	GCAC	1.63%	CTGT	2.02%	GCAT	1.80%	GTAG	2.23%	TTAG	1.75%	TTAA	2.23%	TTAC	2.39%	GCAA	2.01%	TCAT	2.76%	TTGT	3.12%	GTAT	1.57%
13	GCGG	1.64%	GGAG	1.66%	CCGC	1.58%	CCGC	1.80%	ACAG	1.55%	CCAT	2.02%	CCAC	1.68%	ATAG	2.16%	CTGT	1.72%	TTAC	2.02%	TTGA	2.32%	TCAA	1.60%	GCAT	2.68%	CCAT	2.85%	ACAA	1.57%
14	CTGT	1.52%	CTAG	1.65%	CTAG	1.56%	TTAG	1.79%	CCGA	1.54%	GCCT	1.91%	CCAT	1.68%	CTAG	2.06%	CCAC	1.71%	ACAA	1.76%	CTAT	2.00%	CTAG	1.55%	GTAT	2.64%	TCAG	2.33%	ATGT	1.27%
15	TCCT	1.51%	TTGT	1.62%	TCAT	1.55%	CCAT	1.60%	GCCT	1.40%	GCGT	1.81%	GCAC	1.63%	TTЛЛ	2.03%	CTGC	1.59%	ACAC	1.59%	ATЛЛ	1.88%	CCGA	1.52%	TTGA	2.40%	TTAT	2.27%	TTGG	1.13%
16	TTGT	1.44%	TCCT	1.58%	TTGT	1.54%	TTGT	1.48%	GCGT	1.40%	CTGC	1.68%	CTAG	1.60%	TCGA	1.85%	GCGC	1.57%	GCAC	1.59%	CCAC	1.86%	TTGC	1.49%	CCAG	2.03%	ACAA	2.12%	CTAT	1.09%
17	CTGA	1.42%	TCAT	1.58%	TCCT	1.47%	CTGC	1.44%	CTGT	1.30%	GGAG	1.62%	TTAC	1.31%	ATCA	1.77%	GTGT	1.54%	TTAG	1.50%	ACAT	1.84%	TCGA	1.45%	ACAG	1.95%	CTGC	2.06%	CCAT	1.09%
18	TTAG	1.42%	CCGG	1.55%	CTGC	1.43%	GCGG	1.43%	CCGT	1.27%	CCCT	1.47%	ACAT	1.26%	TCAA	1.74%	CTGA	1.53%	TTGA	1.49%	GCAA	1.82%	TCGT	1.43%	CTAT	1.93%	CCAG	2.02%	CTAA	1.04%
19	CTGC	1.40%	CTGC	1.37%	GGAG	1.31%	TCAT	1.36%	CCAT	1.25%	GTGC	1.42%	ATAG	1.19%	ATAT	1.40%	TTAT	1.36%	CCAG	1.40%	GTAA	1.72%	TCAT	1.40%	GCAG	1.92%	CTAT	1.90%	ATGA	1.04%
20	CCGG	1.24%	GCGT	1.35%	GCGG	1.27%	GTAG	1.33%	CTGC	1.25%	GCAT	1.39%	ACAC	1.10%	CCAT	1.28%	TCAA	1.34%	GTAT	1.28%	ATAT	1.67%	ACAG	1.39%	CCAC	1.90%	TCAT	1.81%	GTAA	1.02%

## Data Availability

Not applicable.

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
