# Peer review of "Evolutionary Invariant of the Structure of DNA Double Helix in RNAP II Core Promoters"

_ijms, 2022, doi:10.3390/ijms231810873_

Round 1

Reviewer 1 Report

The authors herein proposed AN interesting paper on the sequence/structural features of core promoter DNA elements in promoters class II. 

The manuscript Is well written and provide interesting insights on the specie specific affinity in the binding of DNA related to variations. 

Overall the main issues are in the experimental details. Materials and methods section Is too poor and a lot of informations are required to be provided so as to test the data.

Reviewer 2 Report

The authors analyzed the sequence and biophysical patterns of the RNAP II core promoters.

1. It is better to explain how the promoter sequences are aligned in EPD briefly.

2. In fig 2, the authors calculated the bits of nucleotides at different sites. I am confused that why some regions don’t have any nucleotides. For example, H. sapiens at -28 position, isn’t the TATA box there?

3. Fig 2 needs to be improved. For example, in the last logo, the letter “T” is truncated.

4. The citation needs to be improved. I guess many others have done this kind of analysis. 

5. Although the authors are trying to find common characteristics among different promoters, how to relate the difference to evolution?

Author Response

We are grateful to you for careful reading of our manuscript and useful comments.

Point 1: It is better to explain how the promoter sequences are aligned in EPD briefly.

Response 1: We added this explanation at the very beginning of the Results section.

Point 2: In fig 2, the authors calculated the bits of nucleotides at different sites. I am confused that why some regions don’t have any nucleotides. For example, H. sapiens at -28 position, isn’t the TATA box there?

Response 2: In this article, we analysed the -50 to +30 region of RNA polymerase II core promoters. There is indeed a TATA box in the -28 region, but it represented by various sequences that can differ from TATAWAAR. The main structural property of these sequences is the ease of bending towards a wide groove. At least 20 of such octanucleotides were found, please see Table 1.

The logo representation in each position depends on the frequency of occurrence of nucleotides in this position and scale parameter. Therefore, the Logo-image of the TATA-box of higher animals does not present some motifs up to the scale 0.1.

Point 3: Fig 2 needs to be improved. For example, in the last logo, the letter “T” is truncated.

Response 3: Thank you, we replaced the truncated figure. Taking into account your remark, we have introduced the Logo-image with information content 1.0 bits into the main text, and the Logo-image with information content 0.4 bits in Supplementary.

We have inserted explanatory text on page 6:

“The logo-representation of promoter sequences with information content 1.0 bits is shown in Figure 2, while that with information content 0.4 bits is shown in Figure S3. We present two options for scaling the logo image to best reveal the features of different fragments of core promoters, because frequencies of occurrence of nucleotides differ sharply in different regions.”

Point 4: The citation needs to be improved. I guess many others have done this kind of analysis. 

Response 4: Almost all works, related to the analysis of Polymerase II core promoters, known to us, have been cited. But taking into account your remark we have added one more citation: [35] Mondal, M., Choudhury, D., Chakrabarti, J., & Bhattacharyya, D. (2015). Role of indirect readout mechanism in TATA box binding protein–DNA interaction. Journal of computer-aided molecular design, 29(3), 283-295 doi:10.1007/s10822-014-9828-x, where molecular dynamics simulation studies of TBP–DNA system was carried out.

Point 5: Although the authors are trying to find common characteristics among different promoters, how to relate the difference to evolution?

Response 5: In our opinion, all the data, presented here, correspond to the “bottom-up” approach conception of evolution, starting from the physicochemical properties of nucleic and amino acid polymers. This concept is presented in the work Auboeuf, D. Physicochemical foundations of life that direct evolution: Chance and natural selection are not evolutionary driving forces. Life2020, 10(2), 7, doi.org/10.3390/life10020007.

We added clarifications in the Materials and Method section, and have done some changes in the English also. All changes in the text are marked in yellow.

Round 2

Reviewer 2 Report

none